# MyD88 TIR domain higher-order assembly interactions revealed by microcrystal electron diffraction and serial femtosecond crystallography

Max T. B. Clabbers[1,10,11], Susannah Holmes[2,11], Timothy W. Muusse[3], Parimala R. Vajjhala[3], Sara J. Thygesen[3], Alpeshkumar K. Malde[4], Dominic J. B. Hunter[3,5,6], Tristan I. Croll[7], Leonie Flueckiger[2], Jeffrey D. Nanson[3], Md. Habibur Rahaman[3], Andrew Aquila[8], Mark S. Hunter[8], Mengning Liang[8], Chun Hong Yoon[8], Jingjing Zhao[1], Nadia A. Zatsepin[2], Brian Abbey[2], Emma Sierecki[5], Yann Gambin[5], Katryn J. Stacey[3,6,9], Connie Darmanin[2✉], Bostjan Kobe[3,6,9✉], Hongyi Xu[1✉] & Thomas Ve[4✉]

MyD88 and MAL are Toll-like receptor (TLR) adaptors that signal to induce pro-inflammatory cytokine production. We previously observed that the TIR domain of MAL (MAL[TIR]) forms filaments in vitro and induces formation of crystalline higher-order assemblies of the MyD88 TIR domain (MyD88[TIR]). These crystals are too small for conventional X-ray crystallography, but are ideally suited to structure determination by microcrystal electron diffraction (MicroED) and serial femtosecond crystallography (SFX). Here, we present MicroED and SFX structures of the MyD88[TIR] assembly, which reveal a two-stranded higher-order assembly arrangement of TIR domains analogous to that seen previously for MAL[TIR]. We demonstrate via mutagenesis that the MyD88[TIR] assembly interfaces are critical for TLR4 signaling in vivo, and we show that MAL promotes unidirectional assembly of MyD88[TIR]. Collectively, our studies provide structural and mechanistic insight into TLR signal transduction and allow a direct comparison of the MicroED and SFX techniques.

[1] Department of Materials and Environmental Chemistry, Stockholm University, Stockholm, Sweden. [2] Australian Research Council Centre of Excellence in Advanced Molecular Imaging, Department of Chemistry and Physics, La Trobe Institute for Molecular Science, La Trobe University, Melbourne, Victoria, Australia. [3] School of Chemistry and Molecular Biosciences, The University of Queensland, Brisbane, Queensland, Australia. [4] Institute for Glycomics, Griffith University, Southport, Queensland, Australia. [5] EMBL Australia Node in Single Molecule Science, University of New South Wales, Kensington, New South Wales, Australia. [6] Institute for Molecular Bioscience, The University of Queensland, Brisbane, Queensland, Australia. [7] Cambridge Institute for Medical Research, University of Cambridge, Cambridge, UK. [8] Linac Coherent Light Source, SLAC National Accelerator Laboratory, Menlo Park, California, USA. [9] Australian Infectious Diseases Research Centre, The University of Queensland, Brisbane, Queensland, Australia. [10]Present address: Department of Biological Chemistry, University of California Los Angeles, Los Angeles, California, USA. [11]These authors contributed equally: Max T. B. Clabbers, Susannah Holmes. ✉email: c.darmanin@latrobe.edu.au; b.kobe@uq.edu.au; hongyi.xu@mmk.su.se; t.ve@griffith.edu.au

Toll-like receptors (TLRs) detect pathogens and endogenous danger-associated molecules, initiating innate immune responses that lead to the production of pro-inflammatory cytokines. Signaling by TLRs is initiated by dimerization of their cytoplasmic TIR (Toll/interleukin-1 receptor [IL-1R]) domains, followed by recruitment of the TIR-containing adaptor proteins, including MyD88 (myeloid differentiation primary response gene 88) and MAL (MyD88 adaptor-like/TIRAP)(Fig. 1)[1]. Combinatorial recruitment of these adaptors via TIR : TIR interactions orchestrates downstream signaling, leading to induction of the pro-inflammatory genes. In previous work, we showed that MAL TIR domains (MAL[TIR]) spontaneously and reversibly form filaments in vitro. They also formed co-filaments with TLR4 TIR domains (TLR4[TIR]) and nucleated the assembly of MyD88[TIR] into crystalline arrays[2]. These results suggested signaling by cooperative assembly formation (SCAF), a mechanism prevalent in innate-immunity and cell-death pathways[3,4], and we proposed a model for signal amplification, in which the TLR4, MAL and MyD88 TIR domains sequentially and cooperatively assemble into a higher-order TIR domain complex. This assembly then induces the formation of the Myddosome, involving the death domains of MyD88 and the protein kinases, IRAK2 and IRAK4, leading to proximity-based activation of these kinases (Fig. 1)[5,6]. The 7 Å cryogenic electron microscopy (cryo-EM) structure of the MAL[TIR] filament revealed a hollow tube composed of 12 two-stranded protofilaments of TIR domains and mutational analyses revealed that protein interactions within these protofilaments are likely to represent higher-order TIR-domain interaction interfaces during in vivo signaling, although the structures formed within cells may be more limited in size[7]. However, the structural basis of how MyD88[TIR] and TLR4[TIR] domains self-assemble and interact with MAL[TIR] remained uncharacterized.

Here we set out to structurally characterize the MyD88[TIR] crystalline assemblies observed in our previous work[2]. As the

crystals were too small for conventional X-ray crystallography, we employed the complementary techniques of microcrystal electron diffraction (MicroED) and serial femtosecond crystallography (SFX). MicroED[8,9] enables structure determination of submicrometre-sized crystals. In MicroED data collection, the crystal is continuously rotated in a transmission electron microscope (TEM)[10–12], analogous to the rotation method used in X-ray crystallography[13], and to related three-dimensional electron diffraction methods in TEM[14]. MicroED can complement existing methods in structural biology such as conventional X-ray crystallography, where growing crystals of sufficient size and crystallinity is often the major barrier to structure determination[15–18]. Indeed, many failed crystallization trials have been shown to contain microcrystals[19–21]. Furthermore, small macromolecular crystals potentially have reduced defects[22–24], and controlled perturbations to the sample, such as soaking and vitrification, may be applied rapidly and more uniformly[24–26]. MicroED has already enabled protein structure determination from microcrystals[9,10,23,27–30], structure solution of a previously uncharacterized metalloenzyme[31], structure determination of membrane proteins from microcrystals embedded in lipidic cubic phase[32–35] and the visualization of ligand-binding interactions[25,36]. Furthermore, MicroED enables the study of biomolecules that naturally aggregate or assemble into microcrystals, facilitating structure determination of several short peptide fragments from thin prion protofibrils[23,37–39]. Such naturally occurring crystalline assemblies are of special interest, as they can reveal the interactions occurring in assemblies within cells, illustrating the underlying mechanisms guiding the assembly formation and providing relevant structural insights.

More or less in parallel to the development of MicroED, SFX has emerged as a powerful technique for structure determination and the study of protein dynamics of microcrystalline samples[24,40–43]. SFX exploits the femtosecond-scale duration of extremely brilliant X-ray free-electron laser (XFEL) pulses for the collection of high-quality diffraction data at room temperature, which occurs before the onset of structure-altering radiation damage[44–47]. In SFX, diffraction data are collected as single snapshots from randomly oriented microcrystals[45,47]. With the crystals delivered to the beam at room temperature and minimal sample handling, challenges associated with cryo-cooling and potential protein conformation restrictions are avoided[48]. SFX has facilitated structure determination from submicrometre crystals of radiation-sensitive proteins[49–51] and membrane proteins such as G protein-coupled receptors[52–55]. SFX has also enabled time-resolved studies of light-sensitive proteins with unprecedented temporal resolution[55–57], enabling the study of reactions initiated by ligand binding and exploiting the submicrometre crystal size for rapid reaction initiation[49–51,54,56–58]. In particular, SFX has advanced fibril studies, e.g., amyloids or microtubules, where the fibrous biomolecule assemblies may have partial or no crystallinity, approaching the regime of single-molecule imaging[59,60].

Here we present MicroED and SFX structures of the MAL-induced MyD88[TIR] microcrystals at 3.0 Å and 2.3 Å resolution, respectively. Importantly, both structures show several distinct remodelled loop regions that adopt conformations that are different from previously determined monomeric X-ray and nuclear magnetic resonance (NMR) structures[61,62]. Crystal packing analysis revealed that the MAL-induced MyD88[TIR] crystals have a two-stranded higher-order assembly arrangement of TIR domains identical to that observed previously within spontaneously formed MAL[TIR] filaments[2], and mutagenesis studies demonstrated that the interfaces within these higher-order MyD88[TIR] assemblies are important for signaling. This identical architecture suggested a unidirectional templating mechanism for

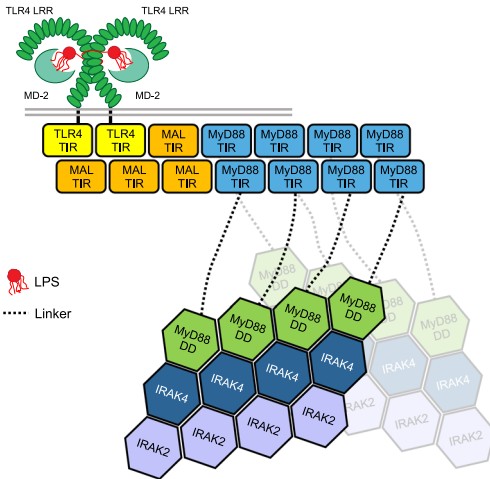

**Fig. 1 Schematic diagram of the SCAF model for TLR signaling.** Pathogen-associated molecular patterns (e.g., LPS) binding to the extracellular LRR domain of a TLR (e.g., TLR4) induces dimerization of its TIR domains, which leads to the recruitment of an adaptor TIR domain (e.g., MAL[TIR]) to the extended surface created by the TLR4[TIR] dimer. Elongation of this trimer through recruiting additional adaptor's TIR domains (e.g., MAL[TIR] or MyD88[TIR]) into a higher-order complex leads to clustering of MyD88 DDs and subsequent recruitment of IRAKs through DD interactions. The initial TIR dimerization and trimerization steps are likely to be unfavourable and rate limiting, whereas subsequent monomer additions are more favourable, rapid and cooperative. LRR, leucine-rich repeat domain; LPS, lipopolysaccharide; TLR, Toll-like receptor; TIR, Toll/interleukin-1 receptor domain; DD, death domain.

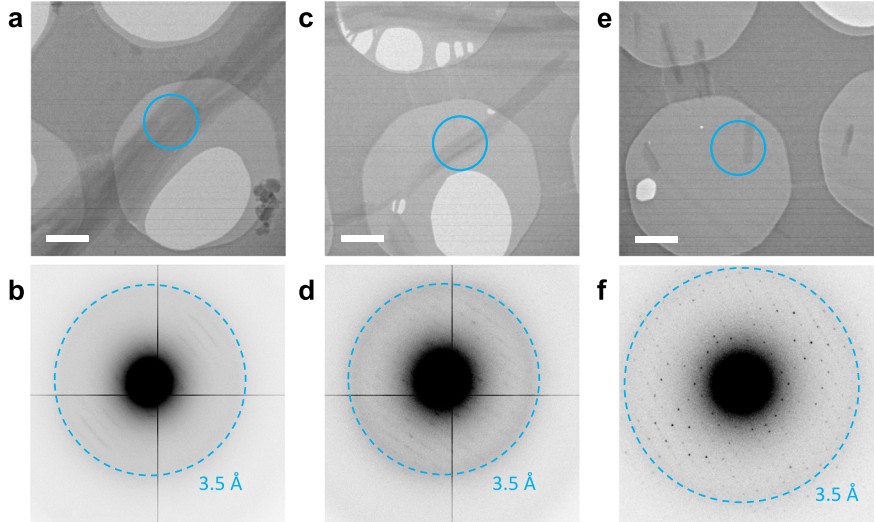

**Fig. 2 MicroED data collection from MyD88$^{TIR}$ microcrystals. a, b** Electron micrograph of aggregated microcrystals, only showing poor-quality diffraction data. Scale bar, 1 μm. **c, d** Multiple microcrystals are overlapping, showing multiple lattices in their corresponding diffraction patterns, complicating data indexing. Scale bar, 1 μm. **e, f** Single hydrated microcrystal, showing high-quality diffraction data up to 3.0 Å resolution. Scale bar, 1 μm. The cyan rings on the micrographs indicate the 1.5 μm diameter parallel beam, defined by the selected area aperture, used for MicroED data collection. Electron diffraction patterns were collected with an angular increment of 0.68° per frame, at a dose rate of 0.12 e⁻/Å² per frame. The data in **a–f** are representative of three EM grids prepared using 3 μl of a 1 : 50 MAL$^{TIR}$ : MyD88$^{TIR}$ crystal solution.

**Table 1 Data collection statistics.**

| Data collection | MicroED[a,b] | SFX[c] |
|---|---|---|
| Temperature (K) | 77 | 300 |
| Space group | C2 | C2 |
| Cell dimensions | | |
| $a, b, c$ (Å) | 99.06, 31.01, 54.30 | 100.40, 31.50. 54.50 |
| $\alpha, \beta, \gamma$ (°) | 90.00, 107.70, 90.00 | 90.00, 107.40, 90.00 |
| Resolution (Å) | 30.54–3.00 (3.11–3.00) | 30.93–2.30 (2.38–2.30) |
| $R_{merge}$ | 0.46 (0.95) | - |
| $R_{split}$ | - | 0.34 (1.3) |
| Mean $I/\sigma(I)$ | 4.8 (1.8) | 2.6 (1.4) |
| $CC_{1/2}$ | 0.95 (0.43) | 0.90 (0.36) |
| $CC^*$ | 0.99 (0.77) | 0.97 (0.73) |
| Completeness (%) | 73.7 (57.3) | 91.4 (60.2) |
| Multiplicity | 12.2 (6.0) | 24.2 (3.4) |

Values in parentheses are for the highest-resolution shell. Intensity statistics were generated from phenix.table_one[98] for the MicroED data and from CrystFel[107,128] for the SFX data.
[a]Merged data from 18 crystals.
[b]MicroED data were truncated at mean $I/\sigma(I) \geq 1.5$ and $CC_{1/2} \geq 0.4$[94].
[c]Merged data from 4725 indexed snapshots out of 13,528 hits.

nucleation and assembly of the higher-order MyD88$^{TIR}$ oligomers, which we confirmed using crystal growth assays. Moreover, structural comparison of the MyD88$^{TIR}$ higher-order assembly and monomeric MyD88$^{TIR}$ enabled us to understand the conformational changes that MyD88$^{TIR}$ monomers undergo upon joining the higher-order assembly. Collectively, our studies shed light on the hierarchical nature of the SCAF mechanism operating in TLR and IL-1R pathways.

## Results
**Data acquisition.** The MAL-induced MyD88$^{TIR}$ microcrystals were typically 100–200 nm in diameter, making them ideally suited to both MicroED (Fig. 2) and SFX.

The microcrystals were deposited on Quantifoil EM grids and vitrified for screening and MicroED data acquisition

(Supplementary Fig. 1). The microcrystals had a tendency to aggregate, forming large bundles that diffracted poorly (Fig. 2a, b). Furthermore, the bent and overlapping crystals complicated the data interpretation (Fig. 2c, d). Using a small parallel electron beam of 1.5 μm diameter, defined by the selected area aperture, only single thin hydrated microcrystals were selected for MicroED data collection (Fig. 2e, f). The MyD88$^{TIR}$ microcrystals diffracted to 3.0 Å resolution and provided high-quality electron diffraction data (Fig. 2f). Data from 18 crystals were integrated, scaled and merged (Table 1). The overall completeness is limited, owing to a preferred orientation of the MyD88$^{TIR}$ microcrystals on the grid and because of the limited tilt range of the goniometer.

The MyD88$^{TIR}$ microcrystals were studied in parallel using SFX. Initially, serial crystallography was attempted on a fixed target at the PETRAIII PII beamline, with a beam size of 2 × 2 μm. However, in this setup, data collection and analysis were complicated by the frequent bundling of microcrystals into larger aggregates (Supplementary Fig. 2). To reach higher resolution and overcome microcrystal aggregation, the sample was delivered as a stream of solvated microcrystals with a gas-dynamic virtual nozzle (GDVN) injector[63,64] to a pulsed XFEL beam at the Linac Coherent Light Source (LCLS), SLAC National Accelerator Laboratory[65]. By using a micro-focused beam (nominally 1 × 1 μm full width at half maximum (FWHM)), and optimizing crystal concentration (7.5 × 10⁸ crystals/ml), high-quality diffraction patterns from individual crystals were collected. Overall, the SFX dataset comprised 4725 indexed patterns from 13,528 hits (35% indexing rate) out of 1,029,868 detector frames (average hit rate of 1.3%). The lattice parameters derived from the SFX data were found to be slightly larger than in the MicroED data collected under cryo-conditions (Table 1).

**Structure solution, model building and refinement.** The structure of MyD88$^{TIR}$ was initially solved using the MicroED data by molecular replacement, finding a well-contrasting unique solution in space group C2. The solution was found with a search model derived from a distantly related Toll-related receptor 2 (TRR2) TIR domain, sharing only 30% sequence identity with

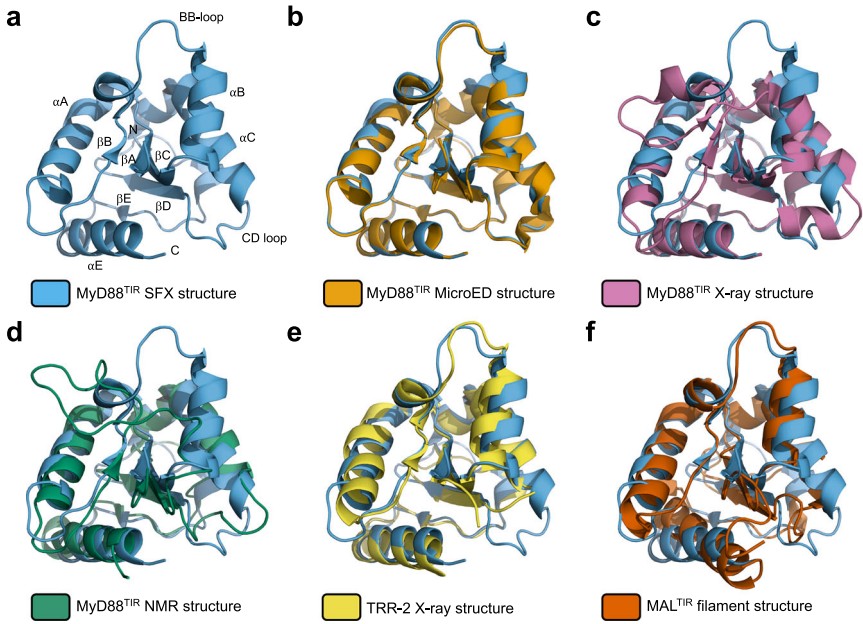

**Fig. 3 MyD88$^{TIR}$ structure comparison. a** Ribbon diagram (blue) of a monomer from the MyD88$^{TIR}$ higher-order assembly structure. Structural elements are labelled sequentially in TIR domains, with the BB-loop connecting strand βB with helix αB, according to the established nomenclature[129]. **b–f** Superposition of the MyD88$^{TIR}$ SFX structure (blue), with **b** the MyD88$^{TIR}$ MicroED structure (orange); **c** the monomeric MyD88$^{TIR}$ X-ray crystal structure (PDB ID 4EO7; magenta); **d** the monomeric MyD88$^{TIR}$ NMR solution structure (PDB ID 2Z5V; green); **e** the crystal structure of the TIR domain of the Toll-related receptor TRR2 from the lower metazoan *Hydra vulgaris* (PDB ID 4W8G; yellow); and **f** the MAL$^{TIR}$ higher-order assembly cryo-EM structure (PDB ID 5UZB; red).

**Table 2 Refinement statistics.**

| Refinement | MicroED | SFX$^a$ | SFX$^b$ |
|---|---|---|---|
| Refinement program | *phenix.refine* | REFMAC5 | *phenix.refine* |
| Resolution (Å) | 30.54–3.00 | 30.94–2.30 | 30.93–2.30 |
| No. reflections | 2436 | 6352 | 6687 |
| $R_{work}/R_{free}$ | 0.223/0.280 | 0.220/0.270 | 0.239/0.281 |
| Mean *B*-factor (Å$^2$) | 52.01 | 40.00 | 45.60 |
| R.M.S. deviations | | | |
| Bond lengths (Å) | 0.005 | 0.002 | 0.001 |
| Bond angles (°) | 0.524 | 1.191 | 0.370 |
| Ramachandran | | | |
| Favoured (%) | 97.79 | 98.53 | 99.26 |
| Allowed (%) | 2.21 | 1.47 | 0.74 |
| Outliers (%) | 0.00 | 0.00 | 0.00 |
| Clashscore | 4.38 | 1.70 | 3.94 |
| Rotamer outliers (%) | 0.00 | 0.76 | 0.00 |

$^a$SFX structure refinement using the REFMAC5 refinement programme.
$^b$SFX structure refinement using the MicroED structure refinement protocol.

MyD88$^{TIR}$ (Fig. 3). The structure of MyD88$^{TIR}$ was iteratively built and refined using the MicroED data (Table 2) and, despite moderate completeness and resolution, the electrostatic potential map showed well-resolved features and enabled remodelling of the loop regions that differed from the previously determined monomeric crystal and solution structures[61,62] (Figs. 3 and 4a). The higher-resolution SFX structure (2.3 Å) was first solved using the MicroED MyD88$^{TIR}$ model as a template for molecular replacement followed by iterative rebuilding and refinement using a different protocol compared to the MicroED structure (Table 2, SFX$^a$). To enable a direct comparison between the MicroED and SFX models, we also solved, rebuilt and refined the SFX

MyD88$^{TIR}$ structure using an identical protocol as described for the MicroED data (Table 2, SFX$^b$). The SFX$^a$ map (Fig. 4b) showed well-resolved features, including water molecules that were not modelled in the MicroED structure. To check whether the MicroED and SFX maps were biased by the search model, simulated annealing (SA) composite omit maps were calculated, confirming the interpretation of our structural models (Supplementary Fig. 3). As the microcrystals contain a small proportion of MAL$^{TIR}$ molecules, there may be a contribution of this heterogeneity to the diffraction, but this is likely to have a negligible effect. Accordingly, there is no evidence of the presence of MAL$^{TIR}$ molecules in the electron density and electrostatic potential maps of the MAL-induced MyD88$^{TIR}$ crystals.

**Structural comparison of MyD88$^{TIR}$ structures.** The MicroED and SFX$^b$ MyD88$^{TIR}$ structures, which were built and refined using the same protocol, are almost identical, with a root mean square deviation (RMSD) of 0.4 Å for 138 Cα atoms. Minor differences in some side-chain conformations can be observed, which is most likely due to the flexibility of certain regions resulting in poorly defined electron density or as a result of the difference in the data collection temperature (Supplementary Fig. 4). The MyD88$^{TIR}$ SFX$^a$ structure was used for the comparison with other TIR domain structures and for the analyses of interaction interfaces within the crystal. The structure of MyD88$^{TIR}$ within the MAL-induced higher-order assembly exhibited conformational differences from the known NMR (RMSD of 2.4 Å for 107 Cα atoms)[61] and X-ray (RMSD of 2.0 Å for 118 Cα atoms)[62] structures of monomeric MyD88$^{TIR}$. This is especially apparent in the region encompassing the BB loop and αB helix, and in the CD loop (Fig. 3). The conformational differences are likely due to participation of these regions in TIR:TIR interactions within the MAL-induced higher-order assemblies. Among the known TIR domain structures, the MAL$^{TIR}$ filament structure (Fig. 3) and the TLR1, TLR2, TLR6 and IL-RACP crystal structures possess similar BB-loop and αB-helix conformations[2].

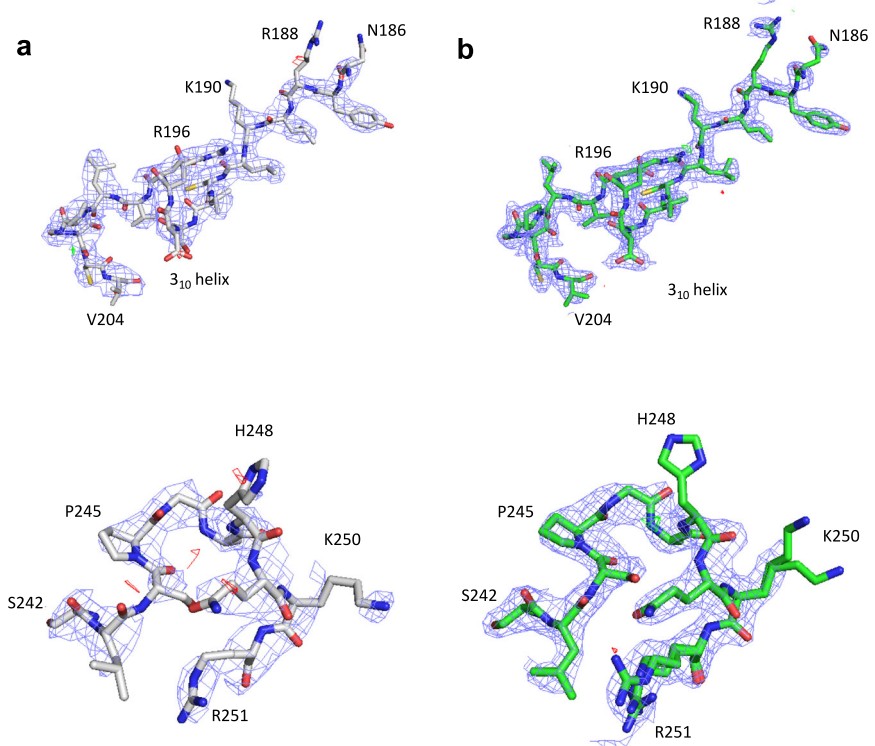

**Fig. 4 Structure determination and model building of the MyD88^TIR higher-order assembly by MicroED and SFX.** Models and maps are presented of the remodelled BB loop (residues 186–204; top) and CD loop (residues 242–251; bottom) for the **a** MicroED and **b** SFX^a structures. The carbon atoms in the MicroED and SFX^a structures are shown in grey and green, respectively. Nitrogen, oxygen and sulfur atoms are shown in blue, red and yellow, respectively. The electrostatic scattering potential (MicroED) and electron density (SFX) 2mFo − DFc maps (blue isomesh) are contoured at 1.2σ, and the difference mFo − DFc maps (green and red isomesh for positive and negative density, respectively) are contoured at 2.8σ. No missing reflections were restored using weighted Fc values for map calculations.

**MyD88^TIR interaction interfaces in the microcrystal.** Analysis of the crystal packing reveals MyD88^TIR higher-order assemblies, each consisting of two offset parallel strands of TIR domains, with subunits in a head-to-tail arrangement forming each strand (Fig. 5a–c and Supplementary Tables 1–3). Formation of the MyD88^TIR assemblies is mediated by two major types of asymmetric TIR domain interactions: one within each of the two strands (intrastrand interface) and one between the two strands (interstrand interface).

Based on the SFX structure, the intrastrand interface involves opposite sides of the MyD88^TIR domain, which together buries ~18.0–18.6% (1500 Å²) of the total surface area per subunit in the structure. It is composed of interactions between residues located in the BB loop of one subunit (BB surface) and the βD and βE strands and the αE helix on the next subunit (EE surface) (Fig. 5b–d and Supplementary Table 1). The highly conserved proline residue (P200 in MyD88) in the BB loop is buried in a shallow pocket between the βE strand and the αE helix consisting of residues I253, C274, L290 and A292. Hydrogen bonds (Supplementary Table 1) and a hydrophobic stacking interaction between the side chains of W284 and R196 stabilize the interface. The conformation of the BB loop is also stabilized by an internal salt-bridge between E183 and R196 (Fig. 5d).

The interstrand interface buries ~12.0–12.2% of the total surface area per subunit (991 Å²) and is composed of interactions between residues located on the αB and αC helices of one molecule (BC surface) and the CD loop and the αD helical region of the partner molecule (CD surface) (Fig. 5b, c, e and Supplementary Table 2). Several residues (W205, F235, K238, F239, L241, P245, I267 and F270) contribute hydrophobic interactions to this interface (Fig. 5e).

The interactions between the MyD88^TIR two-stranded assemblies, which form a continuous sheet in the microcrystals, involve residues predominantly located in the αA helix and the CD and EE loops (Supplementary Table 3). The interface buries ~7–8% (570 Å²) of the total surface area per subunit and is less extensive than the intrastrand and interstrand interactions (Fig. 5f and Supplementary Table 3). These inter-assembly interactions are most likely analogous to non-biological crystal contacts in macromolecular crystals[2,66].

**Mutation of MyD88^TIR intrastrand and interstrand residues perturbs assembly formation and signaling.** We previously showed that alanine mutations of R196, D197, P200, W284 and R288 in the intrastrand interface, and K238, L241, S266 and R269 in the interstrand interface disrupted MAL-induced MyD88^TIR microcrystal formation in solution[2]. To demonstrate the biological importance of the interaction interfaces, we tested the effect of interface residue mutations in a HEK293 TLR4 reporter cell line with an nuclear factor-κB (NF-κB)-driven mScarlet-I reporter and with endogenous *MYD88* knocked out (Fig. 6a, b and Supplementary Fig. 5). Intrastrand mutations R196A, W284A, I253D and R288A abolished NF-κB activation by the TLR4 ligand lipopolysaccharide (LPS), whereas P200A in the BB loop substantially reduced activation (Fig. 6a and Supplementary Fig. 5d). In the interstrand interface, mutants K238A, L241A, F270A and F270E had little or no LPS response. An alanine mutation of F239 in this interface, which predominantly is involved in hydrophobic interactions with αB helix residues within the same subunit, only led to ~20% loss of activity. Mutants localized at the periphery of the interstrand interface had either intact signaling (P245H and R269A)

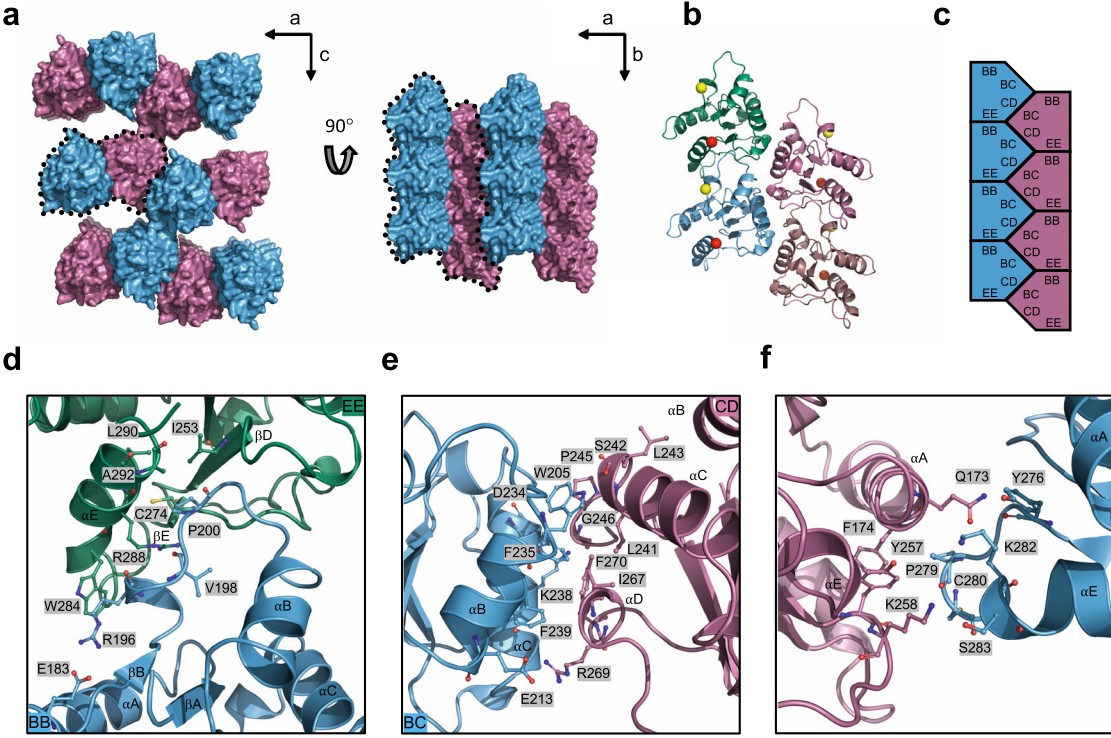

**Fig. 5 Structure of the MyD88^TIR higher-order assembly microcrystal. a** Surface representation of the MyD88^TIR microcrystal, consisting of two-stranded higher-order assemblies (black dotted lines). The two strands are shown in blue and magenta, respectively. **b** Ribbon diagram of the MyD88^TIR higher-order assembly. A yellow sphere indicates the N terminus of each TIR monomer and a red sphere indicates the C terminus of each TIR monomer. The two strands are shown in blue and green, and magenta and dark salmon, respectively. **c** Schematic diagram of the MyD88^TIR microcrystals and the two types of asymmetric interactions within the higher-order assembly. BB surface consist of residues in BB loop; EE surface consist of residues in βD and βE strands, and the αE helix; BC surface consist of residues in αB and αC helices; whereas CD surface consist of residues in CD loop and the αD helical region. **d, e** Detailed interactions within the higher-order assembly **d** intrastrand interface and **e** interstrand interface. **f** Detailed interactions between the two-stranded higher-order assemblies, forming the sheet structure.

or ~20% loss in activity (D234A). Mutant K282A, located at the interface forming the sheet structure that is considered not biologically important (Fig. 5f), also had intact signalling These signaling results agree very closely with analyses of the LPS-induced clustering of expressed MyD88 in cells (Fig. 6b and Supplementary Fig. 5e, f). The results are also consistent with our previous study on spontaneous and MAL-induced MyD88 clustering[2], except that here, using a cell line deficient in endogenous MyD88, an effect of interstrand mutations can be clearly seen.

**Disease-related mutations and post-translational modification sites modulate assembly formation.** Several MyD88 TIR domain missense mutations (V204F, S206C, I207T, S209R, S230N, M219T, L252P and T281P) sustain lymphoma cell survival due to constitutive NF-κB signaling[67–69]. Mapping of these residues onto the MyD88^TIR assembly revealed that the S209R mutation is likely to directly impact interstrand interactions, whereas the T281P mutation may impact intrastrand interactions (Supplementary Fig. 6). L252 is buried and not directly involved in higher-order assembly interactions, but molecular dynamics simulations suggest that this mutation is likely to modulate the conformation of the CD loop[70], which is critical for interstrand interactions in the MyD88 higher-order assembly. To directly test the hypothesis that these disease-related mutations increase MyD88 higher-order assembly formation, we analysed their effects on clustering in both cell-based and cell-free systems (Fig. 6a–c). Consistent with previous reports, expression of the S209R, L252P and T281P mutants in our reporter cell line showed increased basal NF-κB activation (Fig. 6). L252P showed

no further inducibility by LPS, whereas S209R and T281P were LPS responsive. All three mutants had increased basal clustering compared to wild-type (WT) MyD88, which was further increased by LPS for S209R and T281P (Fig. 6b). The aggregation propensity of these mutants was also evaluated by single-molecule spectroscopy, by measuring the brightness of the fluorescence time traces of cell-free expressed green fluorescent protein (GFP)-tagged proteins[71] (Fig. 6c). The S209R and T281P mutants had increased aggregation propensity, forming larger particles than WT MyD88 (Supplementary Fig. 7). By contrast, the L252P mutant formed smaller particles than WT MyD88 (Supplementary Fig. 7), but the complexes were found in higher numbers and formed at lower protein concentrations, as previously reported[72].

The MyD88 TIR domain has been reported to be phosphorylated on S242 (αC helix) and S244 (CD loop), with phosphomimetic mutations of these residues leading to opposite effects on NF-κB activation: the S244D mutation becomes hyperactive, whereas the S242D mutation has an inhibitory effect[70,73]. S242 forms a hydrogen bond with W205 in the MyD88 higher-order assembly and mutation of this residue to an aspartate is thus likely to destabilize the interstrand interface (Supplementary Fig. 6). S244 is not directly involved in higher-order assembly interactions, but similar to L252P, molecular dynamics simulations suggest that the S244D mutation causes a change in the CD loop conformation[70]. When the ability of MyD88 to cluster in HEK293 cells was tested (Fig. 6b), the S244D phosphomimetic mutation increased MyD88^FL clustering, whereas S242D inhibited clustering, which is in perfect agreement with NF-κB activation by these mutants (Fig. 6a). Similar data were

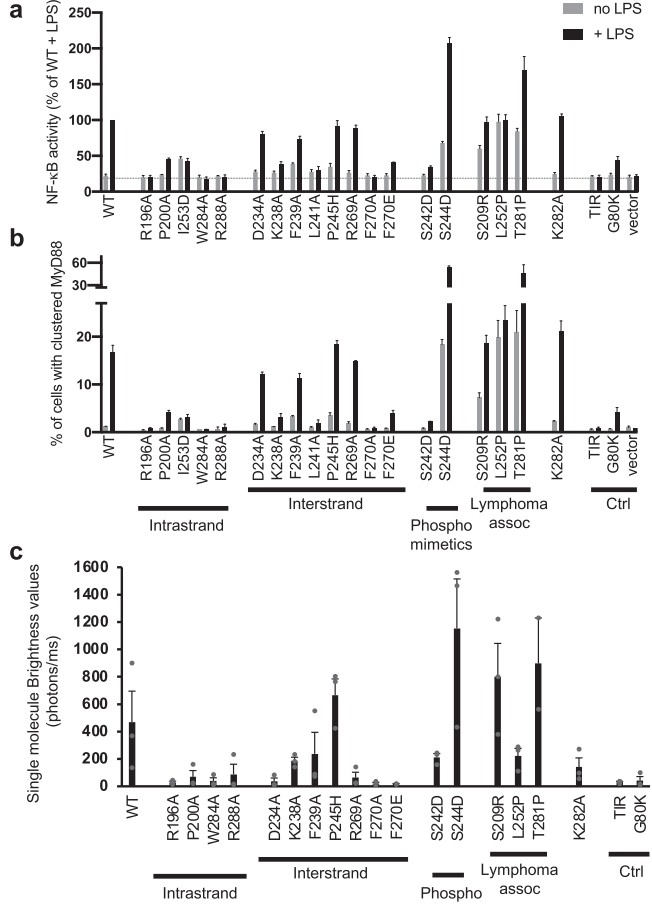

**Fig. 6 Interface, disease-associated and phosphomimetic mutations modulate MyD88 signaling and assembly. a**, **b** Effects of MyD88 mutations on LPS-induced signaling and MyD88 clustering were tested in HEK293 cells expressing TLR4, MD2 and CD14, with *MYD88* knocked out and stably transfected with an NF-κB-driven mScarlet-I fluorescent reporter. The cells were transfected with plasmids expressing wild-type or mutant V5-tagged MyD88, or empty vector, and then treated with (black bars) or without (grey bars) LPS (100 ng/mL) overnight, immunostained to detect MyD88-V5 and analysed by flow cytometry. Cells with very low expression of MyD88 were used for analysis to avoid spontaneous signaling (Supplementary Fig. 5b, c). The mean ± range from $n = 2$ independent experiments is shown. The death-domain mutation G80K, which has previously been shown to prevent MyD88 clustering[130], and a TIR domain alone construct provided negative controls. **a** NF-κB activation measured by the geometric mean fluorescence intensity of the mScarlet-positive population relative to LPS-treated cells expressing wild-type MyD88. The dotted line indicates level of activation in cells with empty vector. **b** The percentage of cells with clustered MyD88 was determined based on the elevated height-to-area ratio of the MyD88 signal, which is observed when MyD88 clusters[2] (Supplementary Fig. 5e). **c** Wild-type MyD88 and mutants were expressed in a cell-free system with an N-terminal GFP tag and the fluorescent samples were analysed by single-molecule spectroscopy on a home-made confocal microscope. To characterize the propensity of wild-type MyD88 and mutants to form higher-order assemblies, the average brightness values (equation (1)) of the proteins were calculated[72]. The results show that S209R, S244D, P245H and T281P mutants have higher propensity than wild-type MyD88 to oligomerize. The mean ± SEM of $n = 3$ or $n = 2$ (F270E and T281P) experiments using different lysate batches with two technical repeats per experiment is shown. The G80K mutant and the TIR domain were used as negative controls.

observed in the single-molecule assay, using cell-free expressed proteins (Fig. 6c). Overall, our new data strongly suggest that MAL-induced MyD88 TIR-domain clustering directly correlates with the level of NF-κB activation and therefore support the relevance of our structure as a model of MyD88 TIR domain association in vivo.

**Comparison of MAL^TIR and MyD88^TIR assemblies**. To gain deeper insights into TIR-domain assembly formation, we compared the MyD88^TIR microcrystal structure (Fig. 5) with our previously published cryo-EM structure of the MAL^TIR filament[2]. Both assemblies share a common overall architecture with head-to-tail intrastrand interactions mediated by the BB and EE surfaces, and interstrand interactions mediated by the BC and CD surfaces (Supplementary Fig. 8a). The conformations of the αE helix and the EE and CD loops are different in MyD88 compared to MAL (Fig. 3 and Supplementary Fig. 8a), resulting in an increase in the buried surface of both the intrastrand and interstrand MyD88^TIR interactions (Supplementary Table 4).

The conformational differences in the αE helix and EE loop also lead to differences in the interface between the two-stranded higher-order assemblies (Supplementary Fig. 8b). In the MyD88-TIR microcrystal, these interactions involve the αA helices and the CD and EE loops, whereas in the MAL^TIR cryo-EM structure the αA, αC and αD helices and the AA and EE loops contribute to these interactions. The differences in these interactions result in distinct packing of the two-stranded higher-order assemblies, MAL^TIR forming a tube consisting of 12 protofilaments, whereas MyD88^TIR forms a continuous sheet (Supplementary Fig. 8c).

**MAL^TIR nucleates MyD88^TIR assembly formation unidirectionally**. MAL^TIR nucleates the assembly of the MyD88^TIR microcrystals[2]. The similar architecture observed in the MAL^TIR and MyD88^TIR higher-order assemblies suggests a molecular-templating mechanism for nucleation and assembly, in which MAL^TIR serves as a platform to promote unidirectional assembly of MyD88^TIR through intra- and interstrand interactions. To test this hypothesis, we captured MyD88^TIR microcrystal growth using differential interference contrast (DIC) and fluorescence microscopy. Either MAL^TIR or GFP-MAL^TIR fusion proteins acted as nucleators of assembly formation and GFP-MAL^TIR nucleates the same type of MyD88^TIR microcrystals as MAL^TIR (Supplementary Fig. 9). Short MAL^TIR-MyD88^TIR crystal seeds were washed to remove MAL and then mixed with MyD88^TIR. The results revealed that MyD88^TIR assembly formation was unidirectional, with a substantial number of seeds observed with growth from one end only (Fig. 7a and Supplementary Movie 1). However, the tendency of MyD88 microcrystals to aggregate also presented a problem here, as the assemblies could also be seen growing in multiple directions from seed aggregates (Supplementary Fig. 10). GFP fluorescence is observed throughout the GFP-MAL^TIR:MyD88^TIR crystal seeds, suggesting MAL^TIR can also incorporate within the MyD88^TIR higher-order assembly, which is consistent with our previous report showing that MAL^TIR and MyD88^TIR can form smaller heterogeneous complex structures when mixed at a 1:1 ratio[2]. As the concentration of GFP-MAL^TIR used for preparing the seeds (0.25–2 μM) is significantly lower than the critical concentration for MAL^TIR filament formation (30 μM)[2], and the initial concentration of MyD88^TIR is ~50–400× higher than GFP-MAL^TIR or MAL^TIR, the seeds must predominantly consist of MyD88^TIR molecules, with a small fraction of MAL^TIR molecules localized at one end and also scattered throughout the seed. Furthermore, the MyD88^TIR assemblies continue to grow after removal of GFP-

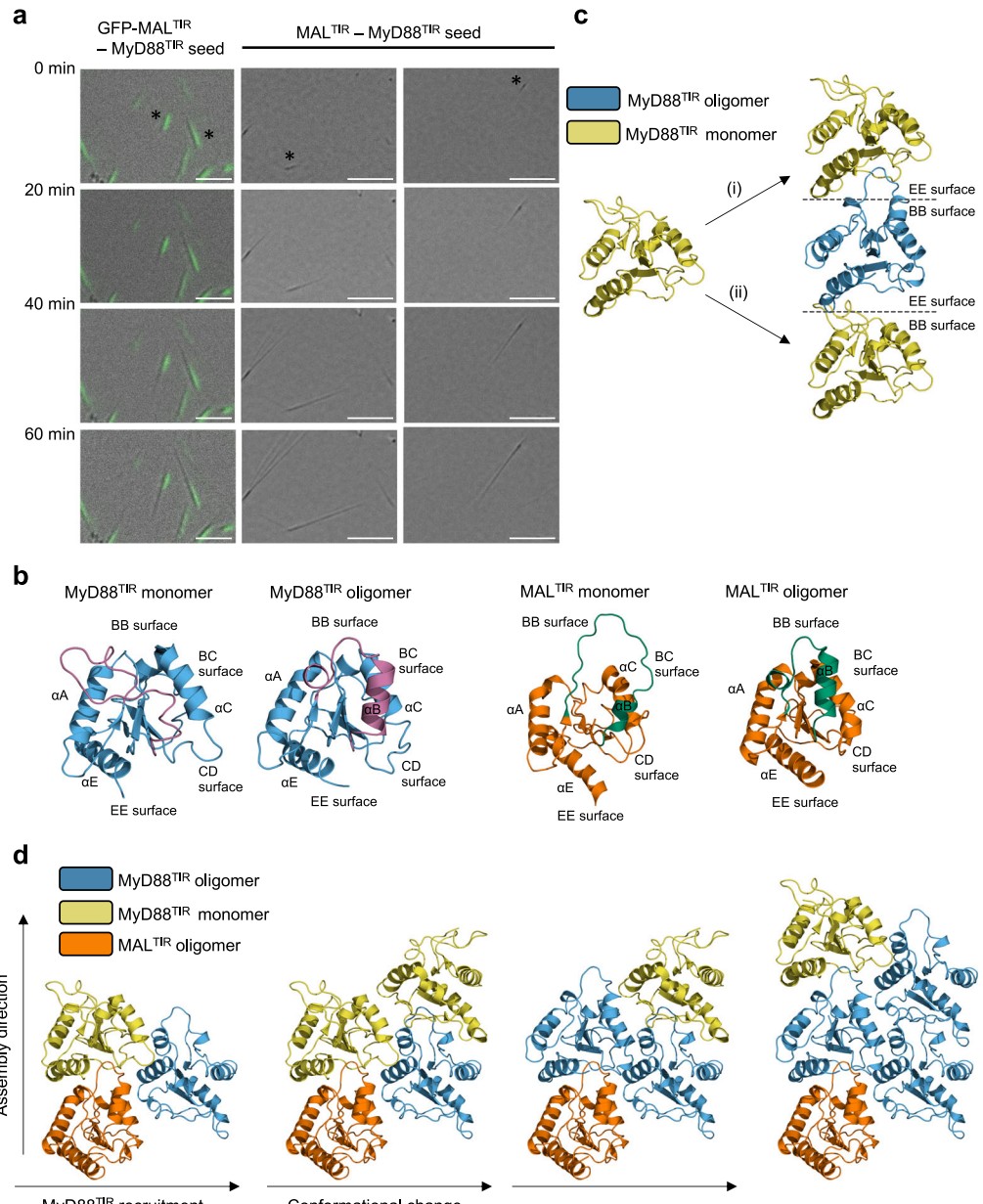

**Fig. 7 MAL^TIR nucleates MyD88^TIR assembly formation unidirectionally. a** Time-lapse imaging of MyD88^TIR microcrystal formation. Representative images of microcrystals growing from single GFP-MAL^TIR-MyD88^TIR and MAL^TIR-MyD88^TIR seeds are shown. The seeds were washed to remove MAL and then mixed with MyD88^TIR. Data are representative of five independent experiments. Asterisks denote seeds with unidirectional growth. Scale bars: left panel 5 µm; middle and right panels 10 µm. **b** Ribbon diagrams of MyD88^TIR (NMR solution structure of monomeric MyD88^TIR (PDB ID 2Z5V) and higher-order assembly structure) and MAL^TIR (NMR solution structure of monomeric MAL^TIR (PDB ID 2NDH) and higher-order assembly cryo-EM structure (PDB 5UZB)), highlighting the rearrangement of the BB loop and αB helix (magenta in MyD88^TIR and green in MAL^TIR) during the monomer-to-oligomer transition. **c** Two models of interstrand interactions, transitioning between MyD88^TIR monomer (yellow) and MyD88^TIR higher-order assembly (blue): (i) EE surface of MyD88^TIR monomer docks onto BB surface of MyD88^TIR higher-order assembly. This interaction does not require any conformational changes in the BB loop and αB helix to occur prior to binding. (ii) BB surface of MyD88^TIR monomer docks onto EE surface of MyD88^TIR higher-order assembly. This interaction requires significant conformational changes in the BB loop and αB helix to occur prior to binding and is therefore less favoured. **d** Model of MyD88^TIR unidirectional assembly formation. The conformational changes in BB loop and αB helix required for the recruitment of new TIR domain subunits are induced by interstrand interactions. The higher-order assembly conformations of MAL^TIR and MyD88^TIR, and the monomeric conformation of MyD88^TIR are shown in orange, blue and yellow, respectively.

MAL^TIR, demonstrating that MAL^TIR is only required for MyD88^TIR assembly nucleation and not elongation.

To predict whether any of the inter- and intrastrand interface surfaces in MAL^TIR are preferred for the interaction with MyD88^TIR, we calculated the predicted buried surface areas of possible MAL^TIR and MyD88^TIR interactions. The calculations

showed that the MAL^TIR BB surface–MyD88^TIR EE surface interaction has the largest buried surface area (Supplementary Table 4). We also mapped the electrostatic potential on the surface of MAL^TIR and MyD88^TIR, and found that the MAL^TIR BB surface and MyD88^TIR EE surface are the only interaction interfaces that are highly charge complementary (Supplementary

Fig. 11). Furthermore, molecular dynamics simulations on MAL$^{TIR}$ : MyD88$^{TIR}$ complexes revealed that complexes involving the MAL BB and MyD88 EE surfaces are more stable than complexes involving the MAL EE and MyD88 BB surfaces (Supplementary Fig. 12). Consistent with these analyses, we have previously demonstrated that mutations in the MAL$^{TIR}$ BB surface prevented full-length MAL-induced MyD88 clustering both in vitro and in cells (R121A, P125A and P125H)[2].

We also compared the structures of MAL$^{TIR}$ and MyD88$^{TIR}$ monomers with their respective structures within higher-order assemblies. This comparison revealed large conformational differences in the BB and BC surface regions (BB loop and αB helix), whereas the EE and CD surface regions adopt similar conformations (Fig. 7b). Models of the recruitment of monomeric MyD88$^{TIR}$ to a growing strand demonstrate that recruitment of new subunits to the assembly via their EE surfaces requires only minimal conformational changes prior to binding, whereas recruitment of new subunits to the assembly via their BB surfaces requires large rearrangements of both the BB loop and αB helix prior to binding, and would therefore be predicted to be less favourable (Fig. 7c). Overall, our structural analyses suggest that in the nucleation and elongation steps of MyD88$^{TIR}$ assembly formation, the EE surface of incoming MyD88$^{TIR}$ molecules dock onto the BB surface of MAL$^{TIR}$ or MyD88$^{TIR}$ subunits. Interstrand interactions via BC and CD surfaces then trigger a rearrangement of the αB helix and BB loop in these newly incorporated TIR domain molecules, enabling them to interact with the EE surface of new incoming MyD88$^{TIR}$ subunits (Fig. 7d).

## Discussion

Over the last decade, crystallography has expanded in several different directions, both in terms of electron crystallography, through developments in MicroED[8,9], and in terms of X-ray crystallography, through SFX[45–47]. Here we used MicroED and SFX to determine the structure of the MyD88 TIR domain from hydrated microcrystalline arrays at 3.0 Å and 2.3 Å resolution, respectively. Both of these techniques have their advantages and disadvantages. SFX utilizes high-intensity X-rays to generate high-resolution structures at room temperature and is able to use injector sample delivery systems to overcome crystal aggregation issues at the expense of high sample consumption (typically 0.3–12 mg of protein)[74–76]. By contrast, MicroED is able to minimize sample consumption (<1 μg protein) allowing for near-complete sampling of reciprocal space using the rotation method of vitrified microcrystals at cryogenic temperature using only a few or even just a single crystal. However, this can come at the expense of often having worse crystallographic quality metrics than is typically achieved in X-ray crystallography. Future advancements in this method, such as serial electron diffraction[77,78], improved electron diffraction detectors, and accurate modelling of the electrostatic potential, taking into account the charged state of atoms and the potential distribution, are likely to improve map quality and provide information about charge interactions[30,79]. For SFX, developments in mix-and-inject experiments at XFELs using nano-focused X-ray beams[80,81] alongside advancements in data analysis[82] will provide future opportunities to conduct time-resolved studies of protein assembly formation. The eventual goal of structural biology at XFELs is to try and push the limits of signal-to-noise, to the point where it is possible to image single molecules in solution[83].

In our investigation, only subtle differences were observed between the MicroED and SFX structures, which may be explained by the differences in the data resolution and completeness, flexibility of certain regions, and difference between cryogenic and room temperature data collection. To our knowledge, only one other group has reported a comparison of these two techniques on the same protein crystal system[24]. Their work showed a slight expansion of the unit cell in the SFX case, which was linked to differences in the data collection temperature. Our room-temperature SFX data also showed a slight increase in lattice parameters along the a-axis, when compared to the cryogenic MicroED data (Table 1), indicating the lattice change is related to the temperature difference between the two data sets.

SCAF, which involves assembly of higher-order oligomers for transmission of receptor activation information to cellular responses, is an emerging theme in signal transduction[4] and operates in several innate-immunity and cell-death pathways including inflammasome signaling[84], RIG-I-like receptor[85] and TLR pathways[2,5]. In this study, we found that the MAL-induced MyD88$^{TIR}$ crystalline assemblies contain a two-stranded head-to-tail arrangement of TIR domains, as previously observed for the TIR domains of the adaptor protein MAL[2]. Analysis of single amino-acid MyD88 mutations for their effect on cellular signaling support the biological relevance of the defined interfaces. Previous functional analyses have measured spontaneous signaling by MyD88 overexpressed in HEK293 cells[70]. Our analysis here has several advantages. First, we used cells with endogenous MYD88 knocked out, which gives a more stringent determination of the function of mutants. This improvement allowed us to demonstrate the importance of residues in the interstrand interactions, which were not apparent in our earlier study[2]. Second, through the use of flow cytometry, we can analyse single-cell responses and select only cells with MyD88 expressed at very low levels to avoid spontaneous signaling. This gives us the ability to observe the response of the mutants to LPS treatment in an intact signaling pathway, avoiding artefacts of overexpression. With this technique, we demonstrated that the R196A mutant is completely inactive, whereas prior work showed it promoted 56% of WT NF-κB activity in the presence of endogenous MyD88, despite having defective TIR domain interactions[70]. Consequently, we are confident in the biological relevance of the signaling assay reported here, which confirmed the importance of several critical residues in both the intra- and interstrand interfaces of the MyD88$^{TIR}$ assembly.

We provide evidence demonstrating that MAL$^{TIR}$ serves as a platform to promote unidirectional assembly of MyD88$^{TIR}$ oligomers. One feature of unidirectional elongation is establishment of hierarchy in the higher-order oligomers, in which upstream molecules can nucleate the assembly formation of downstream molecules, but not vice versa, and appears to be a common feature in many innate-immunity pathways. For example, elongation of the BCL10 adaptor in the CARMA1–BCL10–MALT1 assembly is unidirectional, with growth at one end only as revealed by confocal imaging[86], and structures of the RIG-I : MAVS CARD, the FADD : caspase-8 DED and the MyD88 : IRAK4 : IRAK2 DD assemblies revealed that the RIG-1, FADD and MyD88 oligomers recruit their downstream partners via only one CARD, DD and DED surface, respectively[5,85,87].

Our data add support to a sequential and cooperative mechanism for TLR signal transduction, in which receptor and adaptor TIR domains assemble via the inter- and intrastrand interactions observed in the MyD88$^{TIR}$ and MAL$^{TIR}$ higher-order assemblies, leading to formation of a TIR-domain signalosome. This would then promote clustering of MyD88 DDs to form the Myddosome, with recruitment and activation of IRAKs[5]. The Myddosome defined in vitro is a helical array of DD of MyD88-IRAK4-IRAK2 in a 6 : 4 : 4 arrangement[5]. In contrast to this mechanism suggesting stepwise recruitment of MyD88 proteins, it has recently been proposed that some MyD88 pre-exists in unstimulated cells in a free oligomeric complex via DD interactions, but cannot recruit IRAK4 due to the TIR domain blocking

access to the IRAK4 binding surface[6]. Upon receptor activation, it is proposed that MyD88 TIR domains are recruited into the TLR4$^{TIR}$-MAL$^{TIR}$ signaling complex, releasing the autoinhibition and enabling recruitment of IRAKs to the pre-formed MyD88 oligomer. Further data are needed to validate either of these models, but there are a number of caveats regarding the possibility of pre-formed autoinhibited complexes. First, MyD88 DD surfaces involved in IRAK4 interactions are also required for the assembly of MyD88 DDs into a hexamer and binding of MyD88 TIR domains to these surfaces is likely to prevent DD oligomer formation altogether. Second, there is a sharp concentration dependence for oligomerization of both full-length MyD88 and MyD88 DD in vitro[72] and the threshold for MyD88 clustering in cells is readily exceeded by overexpression. The spontaneous signaling seen with overexpression[88] argues against MyD88 clusters being intrinsically inhibited for IRAK recruitment. At normal cellular concentrations, autoinhibition is likely to play a role in limiting self-association of MyD88[72]. Stepwise TIR domain-mediated recruitment into a TLR signalosome would then increase the local concentration of DD, leading to Myddosome assembly.

In conclusion, our study provides new insights into the architecture and assembly mechanism of TIR-domain signalosomes in TLR pathways, and at the same time allows for a comparison of the complementary techniques of MicroED and SFX. The detailed TIR : TIR interactions reported in this study may also provide templates for designing small-molecule mimics of the important interfaces to inhibit MyD88 higher-order assembly formation for potential therapeutic applications.

## Methods

**Protein production.** Overlapping PCR was used to generate a construct encoding a GFP-MAL$^{TIR}$ fusion protein (EGFP residues 3–239; MAL residues 79–221) with a GSGGS linker, which was cloned into the pMCSG7 expression vector by ligation-independent cloning[89]. For additional information regarding the primers used, please see Supplementary Table 5. MyD88$^{TIR}$ (residues 155–296 in pET28b, C-terminal His$_6$-tag)[2], MAL$^{TIR}$ (residues 79–221 in pMCSG7, N-terminal His$_6$-tag and c-Myc-tag)[2] and GFP-MAL$^{TIR}$ were produced in *Escherichia coli* BL21 (DE3) cells, using auto-induction media[90]. Cells were grown at 30–37 °C until the mid-exponential phase (OD$_{600nm}$ of 0.6–0.8) was reached. The temperature was then reduced to 15–20 °C and the cultures were grown for ~16 h before harvesting. The cells were lysed in 50 mM HEPES (pH 7–8), 500 mM NaCl and 1 mM dithiothreitol, using sonication. The resulting supernatant was applied onto a 5 ml HisTrap FF column (GE Healthcare). The bound protein was eluted using a linear gradient of imidazole from 30 to 250 mM and the fractions containing the protein of interest were pooled, concentrated and applied onto a Superdex 75 HiLoad 26/60 gel-filtration column (GE Healthcare) pre-equilibrated with 10 mM HEPES pH 7.5 and 150 mM NaCl. The peak fractions were pooled, concentrated to a final concentration of 1–10 mg/ml and stored in aliquots at −80 °C.

**MyD88$^{TIR}$ crystallization.** MAL-induced MyD88$^{TIR}$ crystals were produced by incubating MAL$^{TIR}$ (0.5–3 μM) with MyD88$^{TIR}$ domain (60–100 μM) in 10 mM HEPES pH 7.5–8, 150 mM NaCl at 25–37 °C for 60–120 min. GFP-MAL-induced MyD88$^{TIR}$ crystals were produced by incubating GFP-MAL$^{TIR}$ (0.5–3 μM) with MyD88$^{TIR}$ (95 μM) in 10 mM HEPES pH 7.5, 150 mM NaCl at 30 °C for 20–120 min. To produce seeds for crystal growth analysis, the incubation (30 °C) of GFP-MAL$^{TIR}$ (0.5–3 μM) with MyD88$^{TIR}$ (95 μM) in a total volume of 50 μl was stopped after 20 min.

**MyD88$^{TIR}$ crystal growth assays.** GFP-MAL$^{TIR}$-MyD88$^{TIR}$ and MAL$^{TIR}$-MyD88$^{TIR}$ seeds were centrifuged at 2000 × *g* for 5 min and washed three times with 250 μl 10 mM HEPES pH 7.5, 150 mM NaCl. The seeds were resuspended in 100 μl 10 mM HEPES pH 7.5, 150 mM NaCl and diluted 1 : 3200 in the same buffer. Five microlitres of diluted seed was added to the well of an imaging plate (μ-Plate 96 well ibiTreat sterile, Ibidi) with 45 μl MyD88$^{TIR}$ (95 μM). The plate was centrifuged at 1500 × *g* for 5 min and immediately transferred to microscope for imaging. During imaging, the plate was incubated at 30 °C on the Nikon Eclipse Ti2 inverted microscope. DIC and GFP fluorescence images were taken using the ×40 objective lens with ×1.5 magnification.

**MicroED sample preparation and data acquisition.** The MyD88$^{TIR}$ crystal samples were prepared by depositing 3 μl of 1 : 50 MAL:MyD88 microcrystal solution on a Quantifoil 3.5/1.0 (300 mesh) Cu holey carbon EM grid. Excess liquid

was blotted away and the sample was vitrified by flash-cooling in liquid ethane, using a FEI Vitrobot Mark IV (blot force 0, blotting time 6 s). The sample was transferred to a Gatan 914 high-tilt cryo-transfer holder. MicroED data were collected on a JEOL JEM-2100 (LaB6 filament) TEM operated at 200 kV equipped with a Timepix hybrid pixel detector (Amsterdam Scientific Instruments). Screening and MicroED data collection, using the rotation method, was performed via the Instamatic software interface[91]. Diffraction data were collected under parallel beam conditions from an area of ~1.5 μm diameter, defined by a selected area aperture. The sample-to-detector distance was 1830 mm. Data were collected with an exposure time of 1.5–2.0 s and an angular increment of 0.68–0.92° per frame. The electron dose rate applied during data collection was ~0.08 e$^-$/Å$^2$/s. The average tilt range covered per individual crystal was about 30°, corresponding to a total exposure dose of ~5.5 e$^-$/Å$^2$.

**MicroED data processing and structure determination.** Data of 18 crystals were integrated, scaled and merged using XDS[92] and AIMLESS[93] (Table 1). Data were truncated approximately at the average $I/\sigma(I) \geq 1.5$ and CC$_{1/2} \geq 0.4$[94] (Table 1). A distantly related TIR domain homologue, TRR2 from *Hydra vulgaris* (PDB ID 4W8G), was identified as a suitable search model using the automated molecular replacement pipeline MrBUMP[95]. An optimized search model was generated using Sculptor[96]. The structure of MyD88$^{TIR}$ was subsequently solved using molecular replacement in Phaser[97] in the PHENIX software suite[98]. The model was iteratively built and refined using Coot[99], phenix.refine[100] and interactive structure optimization using molecular dynamics in ISOLDE[101]. The model was refined using a 5% test set for $R_{free}$, individual isotropic $B$-factors, electron scattering factors and automated optimization of the data vs. stereochemistry and data vs. ADP (atomic displacement parameters) weighting. The geometry of the structural models was validated using MolProbity[102]. A SA composite omit map was calculated over the entire contents of the unit cell using phenix.composite_omit_map[100], sequentially omitting 5% fractions of the structure. No missing reflections were filled in for map calculations.

**Serial crystallography, PETRAIII synchrotron.** Based on the Roedig et al.[103] design, 2.5 μl of microcrystals in crystallization buffer containing 16% glycerol (3.3 × 10$^8$–1 × 10$^9$ crystals/ml) were deposited on a chip with a pore size of 1 μm (manufactured by Sauna P/L) under a humidified environment. The excess mother liquor was filtered from the crystal by drawing off the solution on the underside of the chip, leaving behind a thin layer of crystals. The chip was then immediately flash frozen in liquid nitrogen and mounted onto the standard goniometer system, under a cryo stream, on the P11 beamline at the PETRAIII synchrotron. The beamline was set up at 12 keV with a beam size of 2 μm and flux measured at 6.4 × 10$^{11}$ p/s. A 100 μm pinhole was used and a capillary beam stop designed by the beamline scientists was incorporated into the beamline. The detector distance was set to 588.3 mm and data were collected using the fly scan mode (exposure of 2 s, step size of 10 μm, oscillation of 0.1° and 2 frames per crystal) on a Pilatus 6 M detector.

**SFX sample preparation and data acquisition.** The crystal concentration tested ranged from 2.5 × 10$^8$ to 2 × 10$^9$ crystals/ml for SFX measurements with the optimal crystal concentration of 7.5 × 10$^8$ crystals/ml. The crystals were filtered through a 20 μm stainless steel filter prior to loading into the sample reservoir for sample injection. SFX data were collected at the coherent X-ray Imaging (CXI) endstation at the LCLS, SLAC National Accelerator Laboratory[65]. A GDVN was used, with an optimized flow rate of 20 μl/min. The XFEL operated at a rate of 120 Hz, delivering 9.6 keV (1.3 Å), X-ray pulses of ~40 fs duration with an estimated 8.3 × 10$^{11}$ photons/pulse at the interaction position, assuming ~50% intensity loss along the beamline. The beam was focused to a diameter of 1 μm FWHM. The data were collected on a Cornell-SLAC Pixel Array Detector[104,105] at a distance of 0.181 and 0.1061 m, for ~111 and 32 min (796,710 and 233,158 detector frames), respectively. The GDVN overcame most of the multi-crystal issues but clogging in the injector lines and in-line filters was an issue. The in-line Peek filters were replaced with 20 μm stainless steel filters and the sample reservoirs were vortexed every 15 min to prevent the crystals from settling. The highest hit rates (~4.5%) were achieved by cycling between delivering sample and washing the sample delivery lines with water every 10 min during data collection. Data were collected from a total of 3.2 ml of crystal solution (~4.3 mg of MyD88$^{TIR}$ mixed with 0.17 mg MAL$^{TIR}$) at room temperature.

**SFX data processing and structure determination.** Hit finding and detector calibration was performed using Cheetah[106] with hit finder 8 and a minimum of 15 peaks per image, and with minimal jet masking used. The CrystFEL software suite[107] was then used for indexing, utilizing MOSFLM[108], XGANDALF[109] and DirAx[110] as indexing algorithms and merged with a partialator (using scaling without partiality modelling), followed by data reduction using AIMLESS[93] in the CCP4 software suite[111].

The SFX MyD88$^{TIR}$ structure was solved, rebuilt and refined using two different protocols, SFX$^a$ and SFX$^b$. SFX$^a$: MyD88$^{TIR}$ structure was solved by molecular replacement using Phaser[98] and a polyalanine model of the MicroED structure as a template. The structure was iteratively rebuilt and refined using Coot[99] and REFMAC5[112] within the CCP4 suite[111]. A 10% $R_{free}$ test set was used for

refinement. The model was first refined using individual isotropic *B*-factors; however, in the final steps of refinement, hydrogens were added to the model and TLS parameters were included to model anisotropic displacements. The geometry of the structural model was validated using MolProbity[102]. SFX[b]: The structure was solved, rebuilt, refined and validated using an identical protocol as described for the MicroED data. A SA composite omit map was calculated using the same protocol as described for the MicroED data.

**Structural analyses**. The Dali[113], PISA[114] and PIC[115] servers and PyMOL (version 2.2.3 Schrödinger, LLC) were used to analyse the structures. Electrostatic potentials were calculated using APBS[116]. Figures were prepared using PyMOL.

**Plasmids and site-directed mutagenesis**. The cDNA encoding luciferase in the NF-κB-driven reporter plasmid (pNFκB-Luc, Stratagene) was replaced with that of the fluorescent protein mScarlet-I (Supplementary Table 5). The resulting plasmid (pNFκB-mScarlet) drives expression of the fluorescent protein, mScarlet-I, upon NF-κB activation. Single point mutations of MyD88 were produced by Genscript in a pEF6-MyD88-V5-His$_6$ plasmid encoding residues 1–296 of human MyD88[2].

**Cell lines and cell culture**. HEK-Blue human TLR4 cells (InvivoGen) were stably transfected with the reporter plasmid pNFκB-mScarlet and a single-cell clone was obtained (HEK-Blue-TLR4-NFκB-mScarlet cell line). The *MYD88* gene was knocked out in this cell line using the CRISPR-Cas9 system and a single-cell clone that did not show any detectable MyD88 expression or LPS response was obtained (HEK-Blue-TLR4-NFκB-mScarlet-MyD88 knockout (KO) cell line). All cells were maintained in Dulbecco's modified Eagle medium with 4.5 g/l glucose, 110 mg/l sodium pyruvate supplemented with Glutamax-1, 10% heat-inactivated fetal bovine serum, 50 U/ml penicillin and 50 μg/ml streptomycin (reagents from Life Technologies). All cells were tested and shown to be mycoplasma-free.

**Evaluation of the effects of MyD88 mutations on higher-order assembly and TLR4 signaling in HEK-Blue-TLR4-NF-κB-mScarlet-MyD88 KO cells by flow cytometry**. To assess the ability of mutant MyD88 to form a higher-order assembly and to restore TLR4 signaling in the HEK-Blue-TLR4-NF-κB-mScarlet-MyD88 KO cell line, 400,000 cells were plated in antibiotic-free media in a 12-well plate and transfected 3–4 h later with 200 ng plasmids expressing WT or mutant MyD88 or empty vector alone, using Lipofectamine 2000 (Thermo Fisher Scientific) according to manfacturer's instructions[2]. After ~16 h, transfection media were replaced with medium with 5% serum and 6–8 h later the cells were treated with or without ultrapure *E. coli* LPS (100 ng/mL; Invivogen) for ~16 h. Cells were collected and fixed for 30 min with 4% paraformaldehyde and immunostained overnight with anti-V5 rabbit monoclonal (D3H8Q) antibody (Cell Signaling Technology) at a 1 : 2000 dilution, followed by goat anti-rabbit–Alexa Fluor-488 (Life Technologies) at a 1 : 10,000 dilution for 1 h[2]. The stained cells were run on a BD Cytoflex S flow cytometer and the data were analysed using the FlowJo software. Cells were first gated to exclude debris on a side scatter vs. forward scatter (FSC) plot and then gated to select single cells on a FSC-width vs. FSC-area plot (Supplementary Fig. 5a). A plot of MyD88-V5 signal vs. mScarlet-I reporter signal showed that the reporter was activated upon TLR4 stimulation with LPS, in cells with MyD88 levels that were below the detection threshold of MyD88-V5 (Supplementary Fig. 5b), and higher levels of expression led to progressively more spontaneous signaling. Thus, cells below the detection threshold of MyD88-V5 were assessed for reporter activation and for the ability of MyD88 to cluster into a higher-order assembly. Reporter activation is expressed as the mScarlet-I mean fluorescence intensity in the mScarlet-I-positive cells or as the percentage of mScarlet-I-positive cells. MyD88 clustering was determined from a plot of MyD88-V5 signal peak height vs. area[2] (Supplementary Fig. 5e). The clustering assay is based on the fact that the signal from detection of a clustered protein, with a fluorescent antibody, results in a fluorescent pulse with increased peak height and decreased width compared to the signal from cells expressing diffuse protein[2].

**Preparation of cell-free extracts**. The *Leishmania tarentolae* Parrot strain was obtained as LEXSY host P10 from Jena Bioscience GmbH, Jena, Germany, and cultured in TBGG media (12 g/L tryptone, 24 g/L yeast extract, 0.8% glycerol, 5.55 mM glucose, 17 mM KH$_2$PO$_4$, 72 mM K$_2$HPO$_4$) containing 0.2% v/v penicillin/streptomycin (Life Technologies) and 0.05% w/v hemin (MP Biomedical)[117]. Cells were collected by centrifugation at 2500 × *g*, washed twice by resuspension in 45 mM HEPES pH 7.6, containing 250 mM sucrose, 100 mM potassium acetate and 3 mM magnesium acetate, and resuspended to 0.25 g cells/g suspension. Cells were placed in a cell disruption vessel (Parr Instruments, USA) and incubated under 7000 kPa nitrogen for 45 min, then lysed by a rapid release of pressure. The lysate was clarified by sequential centrifugation at 10,000 × *g* and 30,000 × *g* and anti-splice leader oligonucleotide was added to 10 mM. The lysate was then desalted into 45 mM HEPES pH 7.6, containing 100 mM potassium acetate and 3 mM magnesium acetate, and snap-frozen until required.

**Protein expression using cell-free extracts**. The MyD88 mutants produced by Genscript in the pEF6-MyD88-V5-His$_6$ vector were Gateway™ cloned into the pCellFree G03 vector to produce *N*-terminally GFP-tagged proteins (Supplementary Table 5)[118]. Cell-free lysates from three different preparations were supplemented with a feeding solution containing nucleotides, amino acids, T7 polymerase, HEPES buffer and a creatine/creatine kinase ATP regeneration system at a ratio of lysate to feed solution of 0.21 and a final Mg$^{2+}$ concentration of 6 mM. Purified plasmid DNA, at a concentration between 100 and 400 ng/mL, was added to the expression reaction at a ratio of 1 : 9 (v/v) and the reaction allowed to proceed for 3 h at 27 °C. Fluorescently tagged expressed proteins were detected by SDS-polyacrylamide gel electrophoresis using a Chemidoc MP imaging system (Bio-Rad, Laboratories Pty Ltd, Gladesville, NSW, Australia). Gels were imaged without further processing, using the inbuilt Alexa 488 (GFP), Alexa 546 (mCherry) and Cy5 (prestained markers) settings to verify expression[117].

**Single-molecule spectroscopy and brightness analysis**. The expressing lysates were diluted 1 in 10 in a buffer containing 50 mM HEPES pH 7.5 and 150 mM NaCl directly in a custom-made 192-well silicone plate. Samples were analysed at room temperature on a Zeiss Axio Observer microscope equipped with a ×40/1.2 NA water-immersion objective (Zeiss C-Apochromat), used to focus a 488 nm laser and collect fluorescence. The fluorescence signal was collected in 1 ms time bins and filtered by a 565 nm dichroic mirror and a 525/50 nm band pass filter optimized for GFP detection[119]. For brightness analysis, the average intensity ($\mu$) and SD ($\sigma$) of the signal were calculated for each 100 s time trace and brightness ($B$) was calculated as $B = \sigma^2 / \mu$[120] (1). For determining the number of large polymers, all expressing lysates were diluted to the same average fluorescence (1000 photons/ms) and raw fluorescence traces were collected as described above, then analysed for the frequency of events of given size. A threshold of 4000 photons/ms was used to discriminate large and small MyD88 assemblies.

**Molecular dynamics simulations**. All molecular dynamics simulations were performed with the MicroED MyD88$^{TIR}$ structure and using the GPU version Gromacs 2019.3[121] on the Gadi cluster at the National Computing Infrastructure, Australia. The Gromos 54A7[122,123] force field was used to model the proteins. Each complex was placed in a truncated octahedron periodic box with a 1.4 nm distance between the protein surface and the edge of the box wall. The protonation state of titratable groups was chosen appropriate to pH 7.0. Each system was simulated under periodic boundary conditions in a rectangular box. The pressure was maintained at 1 bar, by weakly coupling the system to a semi-isotropic pressure bath, using an isothermal compressibility of $4.6 \times 10^{-5}$ bar$^{-1}$ and a coupling constant τP = 1 ps[124]. The temperature of the system was maintained at 298 K by independently coupling the protein–ligand complex, lipids and water to an external temperature bath with a coupling constant τT = 0.1 ps, using a Berendsen thermostat[124]. All bond lengths were constrained using the LINCS algorithm[125]. The Simple-Point Charge[126] water model was used and constrained using the SETTLE algorithm[127]. Each system was energy-minimized for 1000 steps, using the steepest descent method, followed by a position-restrained MD simulation, where all heavy atoms of protein were restrained to their original position using 1000 kJ/mol/nm$^2$, allowing water molecules to equilibrate. The restraints were removed and the whole system was allowed to equilibrate for 5 ns. The MD simulations were performed for 100 ns in duplicate, starting with different initial velocity distribution for each system. All coordinates, velocities, forces and energies were saved every 10,000 steps for analysis. The stability of the protein and protein–ligand complexes was evaluated, by measuring the RMSD of protein backbone atoms by fitting the backbone atoms of protein and comparing the initial and final structures.

**Reporting summary**. Further information on research design is available in the Nature Research Reporting Summary linked to this article.

## Data availability

Data supporting the findings of this manuscript are available from the corresponding authors upon reasonable request. A reporting summary for this article is available as a Supplementary Information file. The atomic coordinates and structure factors of the MicroED and high-resolution SFX models (SFX[a] and SFX[b]) have been deposited in the Protein Data Bank under accession codes 7BEQ, 7L6W and 7BER, respectively. Raw MicroED data are available from the SBGrid Data Bank (doi:10.15785/SBGRID/814). SFX data are available at CXIB.org (https://doi.org/10.11577/1767965). Source data are provided with this paper.

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

## Acknowledgements

We thank Ronan Keegan and Tim Gruene for insightful discussions on molecular replacement and model building. We thank Xiaodong Zou (X.Z.) for insightful discussions and critical manuscript reading. We acknowledge the use of the Centre for Microscopy and Microanalysis at the University of Queensland. The work was supported by the Swedish Research Council (2017–05333 to H.X., 2019–00815 to X.Z.), the Knut and Alice Wallenberg Foundation (2018.0237 to X.Z.), the SciLifeLab technology development project (MicroED@SciLifeLab to H.X.), the Wellcome Trust (209407/Z/17/Z supporting T.I.C.), the National Health and Medical Research Council (NHMRC grants 1107804 and 1160570 to B.K. and T.V., 1071659 to B.K. and 1108859 to T.V.) and the Australian Research Council (ARC) Laureate Fellowship (FL180100109 to B.K.). T.V. received ARC DECRA (DE170100783) funding. C.D., B.A., N.A.Z. and S.H. acknowledge the support of the Australian Research Council through the Centre of Excellence in Advanced Molecular Imaging (CE140100011). This work was supported by computational resources provided by the Australian Government through NCI-Gadi under the National Computational Merit Allocation Scheme (Project cj47) and Queensland Cyber Infrastructure Foundation (QCIF, Project fi49). Portions of this research were carried out at the Linac Coherent Light Source (LCLS), a National User Facility operated by Stanford University on behalf of the U.S. Department of Energy, Office of Basic Energy Sciences. The CXI instrument was funded by the LCLS Ultrafast Science Instruments (LUSI) project funded by the U.S. Department of Energy, Office of Basic Energy Sciences. Use of the LCLS, SLAC National Accelerator Laboratory, is supported by the U.S. Department of Energy, Office of Science, Office of Basic Energy Sciences under Contract number DE-AC02-76SF00515. Portions of this research were carried out at the light source PETRAII, P11 beamline at DESY, a member of the Helmholtz Association (HGF). We thank Dr. Anja Burkhardt, Dr. Alke Meents and Dr. Olga Lorbeer for assistance with using the beamline P11.

## Author contributions

T.V., B.K., H.X. and C.D. conceived the project and provided project leadership. T.V. designed the crystallization experiments. T.V., J.D.N., C.D. and S.J.T. produced crystals. S.J.T. and K.J.S. designed and performed crystal growth experiments and analysed the data. T.V., J.D.N. and M.H.R. expressed and purified proteins. H.X. designed MicroED experiments. H.X., J.Z. and M.T.B.C. were involved in MicroED sample preparation, screening and data collection. M.T.B.C. and H.X. performed analysis and structure determination using the MicroED data. T.I.C. performed model building. C.D. designed and led SFX experiments. C.D., L.F., A.A., M.S.H. and M.L. collected SFX data. S.H., N.A.Z., C.H.Y., B.A. and C.D. analysed SFX data. T.V., M.T.B.C. and S.H. analysed the MicroED and SFX structures. A.K.M. designed and performed the molecular dynamics experiments and analysed the data. T.V. designed site-directed mutants. K.J.S, P.R.V. and T.W.M. designed and performed cell biology experiments, and analysed the data. E.S., Y.G. and D.J.B.H. designed and performed single-molecule spectroscopy experiments, and analysed the data. T.V., B.K., C.D., H.X., S.H. and M.T.B.C. wrote the initial draft. All authors contributed to editing and writing the paper.

## Competing interests

The authors declare no competing interests.
