## [Peer Review File · Nature Communications]

REVIEWER COMMENTS

Reviewer #1 (Remarks to the Author):

In the article “MyD88 TIR domain high-order assembly interactions revealed by microcrystal electron diffraction and serial femtosecond crystallography,” the authors report a molecular mechanism on how the MyD88 TIR domain oligomerizes into a fibril assembly by two different structural biology techniques. The cellular machineries coupled to membrane-bound receptors are often regarded as individual functional molecules, while recent studies indicate that many can cluster into homo- or hetero-oligomers that play important roles in cell signaling processes. The authors focus on MyD88 and MAL, two Toll-like receptor adaptor proteins in this study, to dissect their oligomerization mechanisms. By applying MicroED and SFX, they overcame the crystal size limitation of MyD88 and solved two high-resolution structures. These structures indicate a novel interaction network allowing MyD88’s TIR domain to form fibrils nucleated by MAL crystals. The authors further tested the functional effects of the residues that are responsible for the oligomerization. Finally, by comparing the monomeric and oligomeric structures of MyD88 TIR and MAL TIR, a model of MyD88 assembly that is initiated and recruited by MAL oligomers is proposed. This study therefore provides direct evidence of a TIR domain from different proteins assembling into ordered oligomers, which can help this field understand how membrane signals transmit intracellularly. In addition, this study is the second to compare two modern protein microcrystallography techniques, MicroED and SFX, using the same target protein; this further validates both as complementary structural biology methods. I recommend publication in Nature Communications after the following points are addressed:

1. Density maps from either MicroED or SFX do not strongly support sidechain rotamers of a few residues. One example is H248 in Figure 4 (and a few more in the supplementary figures), so the indication for H248’s rotameric change between two structures is not solid evidence (Figure S4c), although it is reasonable to suggest that flexibility may cause the unresolved density.
2. The crystal growth assay indicates that MAL mainly exists in the nucleation of MyD88 crystals and only partially incorporates into the MyD88 crystals. Due to the tendency for the filamentous crystals to aggregate (Figure 2a-b) and for long filamentous growths to produce poor diffraction (Figure 2c-d), small thin crystals were selected for MicroED analysis (Figure 2e-f). These small crystals would likely contain the MAL nucleation point and therefore the resulting diffraction could be from both MAL and MyD88 even though the ratio of MAL:MyD88 is 1:50 in the MicroED experiment. In addition, MicroED data is often collected at the tip or edge of a crystal in order to obtain maximal diffraction signal. The authors should clarify the source of the diffraction in these crystals – whether they are indeed solely of MyD88, or if there is mixed signal including MAL.
3. Macromolecular crystallography experiments generally require protein concentration to exceed physiological parameters. It would be of great interest to show how well these two proteins interact at lower concentrations as a meaningful control experiment.
4. It was mentioned in the Results and Discussion sections that subtle differences in the MicroED vs. SFX structures (i.e., the unit cell parameters) may be related to the difference in data collection temperature, in addition to differences in data resolution/completeness and

flexibility of certain residues. Would it be feasible to collect MicroED data of these crystals at room temperature (with a similar radiation damage profile as data collected in cryo, and without excess crystal aggregation) to provide a closer comparison to the SFX data?

5. Figure 7a lacks scale bars in the micrographs.

Reviewer #2 (Remarks to the Author):

This manuscript describes the higher order assembly of the MyD88 TIR domain using microED (at cryogenic temperature) and SFX (at room temperature) techniques, two methods suitable for microcrystalline materials. The authors previously published the MAL TIR domain filaments that induce the formation of higher order MyD88 TIR domain microcrystals. Here they show that similar to MAL TIR domain assembly, the MyD88 TIR domain forms a two-stranded arrangement of the TIR domain assembly induced by the MAL TIR domain through unidirectional templating. Comparison with previously reported monomeric MyD88 TIR domain structures reveals conformational changes in the BB loop/aB helix and CD loops of the TIR domain that are conducive to the higher order assembly. The MyD88 TIR domain assembly interfaces observed are essential for the TLR signal transduction, which predominantly employs MyD88 as a central adapter molecule. The structures of the MyD88 TIR domain assembly have been further probed and validated through a number of biochemical analyses, cellular assays, and molecular dynamics simulations. The authors further provide molecular mechanisms for the disease-inducing mutations in MyD88 at or near the observed interfaces. As such, this manuscript should appeal to the readership of Nature Communications.

There are only minor issues to be addressed before publication.

Figure 4 and Figure S4 seem to provide redundant information about the fit of the models and experimental data from microED/SFX.

For some reason hydrogen bonds were not shown in figure 5 but were shown in figure S6.

Figure 6 appears to show that mutations at the intrastrand interface led to more severe disruption of signaling than interstrand mutations. Does this suggest that a single strand of MyD88 TIR may be sufficient for signaling in cells, as the concentrations of MyD88 in cells may be more limited than the in vitro assays the authors used?

Figure S8C, can the authors put the MAL TIR domain circular protofilaments, the MyD88 TIR domain "sheet", together with a TIR dimer from a TLR into one consistent model of the TIR domain signaling complex?

Reviewer #3 (Remarks to the Author):

This submission follows up previous studies of the Toll-like receptor signalling adaptors Mal and MyD88 in which they found that Mal nucleated the assembly of MyD88 TIR into filaments and rod-like assemblies. Here they have found conditions where Mal can seed the formation of MyD88 TIR microcrystals and have solved the structures to molecular resolution

using two cutting edge techniques, microcrystal electron diffraction and serial femtosecond crystallography. This seems important from a methodological point of view because these techniques could find wide use where only very small (micron) sized crystals are available.

The monomer structures solved are closely similar to those already solved by X-ray crystallography and NMR although there are some differences for example in the BB-loop that are reflective of TIR-TIR interactions in the microcrystals. The MyD88 TIRs form two stranded structures within the microcrystal stabilised by inter and intra strand TIR-TIR interactions. The key question then is the extent to which these higher order structures are relevant to physiological signalling. To address this the authors have carried out extensive site directed mutagenesis of the interface residues observed in the microcrystals and test these mutants in cell culture signalling assays. A subset of these including residues such as P200 known to be critical for signalling have impaired responses to LPS. These results are corroborated by in cell clustering and single molecule brightness studies. Together this provides substantial evidence that the TIR-TIR interactions observed in the microcrystal assemblies or subsets of them play a critical role in signalling in vivo. These assays were also used for mutants associated with lymphomas and two phosphor-sites which validate previous studies.

In a final set of experiments the authors have studied the growth of the Mal seeded filaments/assemblies and find that growth of these is unidirectional. They provide a nice explanation for this observation, that the architecture of the Mal BB loop prevents addition of subunits at this end.

Overall the studies presented in this paper are of high quality and provide important new insights into the molecular mechanisms of signal transduction through the TLR pathway, and will be of wide interest to those working on the molecular mechanisms of signal transduction.
Nick Gay

We thank the reviewers for their very positive reviews. We are thrilled that they appreciate the quality and significance of our study. Reviewer 1 and 2 had a number of comments that we have responded to below.

Reviewer #1:

Comment 1: Density maps from either MicroED or SFX do not strongly support sidechain rotamers of a few residues. One example is H248 in Fig. 4 (and a few more in the supplementary Figures), so the indication for H248's rotameric change between two structures is not solid evidence (Fig. S4c), although it is reasonable to suggest that flexibility may cause the unresolved density.

Response: The regions highlighted in Fig. 4 and Supplementary Fig. 4 have been modelled as best we can based on the MicroED and SFX density maps. We agree with the reviewer that the unresolved densities are likely due to flexibility and we have stated this more explicitly in the 'Structural comparison of monomeric and higher-order assembly MyD88^{TIR} structures' section of the revised manuscript.

Comment 2: The crystal growth assay indicates that MAL mainly exists in the nucleation of MyD88 crystals and only partially incorporates into the MyD88 crystals. Due to the tendency for the filamentous crystals to aggregate (Fig. 2a-b) and for long filamentous growths to produce poor diffraction (Fig. 2c-d), small thin crystals were selected for MicroED analysis (Fig. 2e-f). These small crystals would likely contain the MAL nucleation point and therefore the resulting diffraction could be from both MAL and MyD88 even though the ratio of MAL:MyD88 is 1:50 in the MicroED experiment. In addition, MicroED data is often collected at the tip or edge of a crystal in order to obtain maximal diffraction signal. The authors should clarify the source of the diffraction in these crystals – whether they are indeed solely of MyD88, or if there is mixed signal including MAL.

Response: In this study we did not choose specific parts of the crystals to collect MicroED data from. Instead, data was collected on different parts of the crystals, since the needle-shaped crystals are thin across their entire length. Since our crystal growth assays (Fig. 7a, Supplementary Fig. 10 and Supplementary Video 1) demonstrate that MAL molecules are likely to exist in the nucleation site and also are scattered throughout the crystals - MAL may have contributed to the diffraction observed in our MicroED and SFX experiments. Although we cannot rule out a mixed MyD88/MAL diffraction signal, there is no evidence of the presence of MAL molecules (30% sequence identity with MyD88) in the electrostatic potential (MicroED) or electron density (SFX) maps of the MAL-seeded MyD88 crystals. The contribution to the diffraction signal from MAL molecules in these crystals must therefore be negligible. To provide clarification about the source of diffraction we have included the following statement at the end of 'Structure solution, model building and refinement' section in the revised manuscript:

‘As the microcrystals contain a small proportion of MAL^{TIR} molecules, there may be a contribution of this heterogeneity to the diffraction, but this is likely to have a negligible effect. Accordingly, there is no evidence of the presence of MAL^{TIR} molecules in the electron density and electrostatic potential maps of the MAL induced MyD88^{TIR} crystals.’

Comment 3: Macromolecular crystallography experiments generally require protein concentration to exceed physiological parameters. It would be of great interest to show how well these two proteins interact at lower concentrations as a meaningful control experiment.

Response: We do not anticipate that MAL and MyD88 will associate to an appreciable level at physiological concentration, as that would lead to spontaneous signalling in vivo. The assembly of a MAL seed for recruitment of MyD88 should require association with a TLR4 dimer. Future work could employ single molecule analysis of the concentration at which MAL and MyD88 oligomerisation is triggered, but this is anticipated to be higher than physiological in the absence of TLR4.

Comment 4: It was mentioned in the Results and Discussion sections that subtle differences in the MicroED vs. SFX structures (i.e., the unit cell parameters) may be related to the difference in data collection temperature, in addition to differences in data resolution/completeness and flexibility of certain residues. Would it be feasible to collect MicroED data of these crystals at room temperature (with a similar radiation damage profile as data collected in cryo, and without excess crystal aggregation) to provide a closer comparison to the SFX data?

Response: We agree with the reviewer that room temperature MicroED experiments would be a great addition to structural biology. However, currently MicroED data collection at room temperature isn't feasible for macromolecular samples. Significant breakthrough in methodology is required before we could preserve hydrated protein crystals in a TEM, for example with the use of in-situ liquid cell. Furthermore, increased radiation damage at room temperature would also limit the data quality.

Comment 5: Fig. 7a lacks scale bars in the micrographs.

Response: We have added scale bars to Fig. 7a, and the related Supplementary Fig.10 and Supplementary Video 1.

Reviewer #2

Comment 1: Fig. 4 and Fig. S4 seem to provide redundant information about the fit of the models and experimental data from MicroED/SFX.

Response: We agree with the reviewer that Supplementary Fig. 4c and Fig. 4 provide redundant information and we have therefore removed Supplementary Fig. 4c from the revised manuscript. Supplementary Fig. 4a-b and d, however, show different regions in the MicroED and SFX structures compared to Fig. 4 and we have kept them in the revised manuscript.

Comment 2: For some reason hydrogen bonds were not shown in Fig. 5 but were shown in Fig. S6.

Response: Addition of hydrogen bonds will make Fig. 5 more crowded and difficult to interpret. To maintain consistent style between these figures, we have removed the dotted line indicating a hydrogen bond between S242 and W205 in Supplementary Fig. 6.

Comment 3: Fig. 6 appears to show that mutations at the intrastrand interface led to more severe disruption of signaling than interstrand mutations. Does this suggest that a single strand of MyD88 TIR may be sufficient for signaling in cells, as the concentrations of MyD88 in cells may be more limited than the in vitro assays the authors used?

Response: Some of the mutations in the interstrand interface (K238A, L241A, and F270A) prevented signalling, just like the intrastrand mutations. This indicates the two stranded structure is likely to be essential in cellular signalling. As stated in the text, the other mutants that had little or no effect on signalling were more peripheral in the interface or predominantly involved in stabilising the structure of the individual subunits.

Comment 4: Fig. S8C, can the authors put the MAL TIR domain circular protofilaments, the MyD88 TIR domain “sheet”, together with a TIR dimer from a TLR into one consistent model of the TIR domain signaling complex?

Response: We are currently preparing another paper which contains the data on which to base a complete model of TLR4-MAL-MyD88 interactions. It will only confuse the situation to present a prediction at this stage, other than what is shown in Fig. 1. However, please note that as stated in the text, the circular filament and sheet structure shown in Supplementary Fig. 8c are not part of the proposed signalosome, which only involves the two stranded protofilament (shown in Fig. 5).